# Design and Simulations of a Self-Assembling Autonomous Vertical Farm for Urban Farming

**Bhanu Watawana** *,† and **Mats Isaksson** †

Department of Mechanical Engineering and Product Design Engineering, Swinburne University of Technology, John St, Hawthorn, VIC 3122, Australia
* Correspondence: bwatawana@swin.edu.au
† These authors contributed equally to this work.

**Abstract:** Vertical farming has been proposed as a solution for diminishing arable land as it significantly reduces the footprint of farming. Most contemporary solutions use a low level of automation; however, automation of vertical farming is currently attracting attention from researchers. This paper introduces a conceptual design for an autonomous vertical farm where the main novelty is the self-assembling feature. The proposed system is designed to be installed and used by a non-specialist. The system is designed for cost minimisation, using one set of resources moved by a robot arm to service the plants. These resources include sensors, a depth camera, and the required farming tools. The farm has the capacity of self-powering, greenhouse conversion, data sharing and learning, and several other features. The paper provides the conceptual design in addition to an analysis of the dimensioning of the robot arm, time studies for operation, and an analysis of the self-powering ability.

**Keywords:** backyard farming; self-assembly; vertical expansion; factory in a box; automated farming; vertical farming; sustainable farm





## 1. Introduction

The diminishing availability of farmable land per capita is a global issue. The global population is projected to reach 9.8 billion by 2050 [1]. According to population growth statistics from the United Nations Food and Agriculture Organisation (FAO) [2], by 2050, the arable land per capita is anticipated to diminish to 33% of the availability in 1970. A joint study conducted by the University of Melbourne and Deakin University [3] concluded that due to shortage of arable land and water and an increasing population, Melbourne's current foodbowl capacity of 41% of Melbourne's total food requirement will be reduced to 18% by 2050. The loss of farmable land is mainly due to climatic changes, urbanisation, desertification, and unavailability of fresh water [4]. The increasing consumer demand for freshness in food products is hampered by transport logistics as farming lands are often situated in distant rural areas. An increasing number of people view the use of artificial fertilisers and pesticides to be a health issue, changing consumer demand focus towards organically grown food. The home garden concept promoted in developing countries [5] caters to the quality requirements but not the quantity. Additionally, the majority of people in developed countries are unlikely to have the time or interest in managing their own farms. The introduction of a low-cost autonomous platform for robotic farming could support the increase in household farming. As a result of the high demand for fresh greens in the urban environment, solutions, such as portable farming inside containers [6], personal food computers [7], and household hydroponic farms [8], are currently being introduced.

Vertical farming is a concept aiming to increase the amount of arable land by 'growing upwards' [9]. The concept can be explained by a single tall glasshouse with multiple layers of racks for growing crops. This is an extension of a hydroponic greenhouse, addressing the lack of arable land and soil, and aiming to reduce the use of herbicides and pesticides [10].

Vertical farming can be implemented in an urban environment, providing the urban community access to fresh food. However, the sustainability of vertical farming is in question as the cost of production is considerably high [11]. One of the main costs is the requirement for artificial lighting. Lighting within a greenhouse makes up 30–50% of the operation costs [12]. Another drawback is that the plants in the system are typically interconnected, sharing water, nutrients and air. Hence, the risk of losing the entire yield due to a disease or malfunction of the system is considerable [13,14].

Automation of vertical farming can provide automated or semi-automated operations within the vertical farm, including automated monitoring and maintenance of the system. Vertical farming automation can be analysed in five stages, basic growth automation, conveyor automation, adaptive automation, system automation, and full automation [15]. Adaptive automation, system automation, and full automation are still being researched [15].

Ease of use and installation are crucial features for a household farming solution. The self-assembling vertical farm solution presented in this paper can be considered as the first step towards a 'Robot-based self-assembling mobile factory', a novel concept, where the factory can assemble itself, automate the process with the same resources used for assembly, and when required, dismantle and pack itself for transportation. Concepts such as 'factory-in-a-box' [16] and 'mobile manufacturing system' [17,18] have been discussed in academia since 2007. Self-assembly is defined as the autonomous organisation of components into a pattern or structure without human intervention [19]. Self-assembly in a multi-robot system allows robots to physically connect to each other to create a distinctive collective robot morphology. Autonomous self-assembly is still uncommon, and most of the current multi-robot systems have pre-programmed morphology to conduct specific tasks [20]. Multi-robot systems or collective robot systems can be categorised into two specific groups, second-order robotics and swarm robotics. A robotic system that can be physically connected to another robot system is considered a second-order system. For a second-order system, the two main classes are self-configurable robots and self-assembling robots [21,22]. Self-configurable robots are composed of modules with little or no independent mobility and only a few sensors, where the modules can connect to each other to form a complex mechanism [23]. A robotic system with the ability to move and operate independently and connect to one another to perform a task that needs collaborative action can be identified as a self-assembling robotic system. The main limitation of a self-assembling robotic system is the lack of full autonomy at the level of robotic modules [21].

Self-assembly, in the context of the mechanism presented in this paper, refers to a mechanism able to assemble itself to conduct specific operations. Examples of such mechanisms include tower cranes, a robot on a track where the robot lays its own track, a multi-arm robot where some arms can be used to modify the other arms. A climbing tower crane uses a mechanism to increase the height of the crane using a prefabricated module structure known as a mast. The climbing frame jacks the slewing structure from the topmost mast and inserts a mast replacing the climbing frame to increase the height [24]. The expansion mechanism of a tower crane is straightforward; however, in contrast to the mechanism proposed in this paper, a tower crane is not designed to access its own base structure.

This paper introduces a conceptual design of a self-assembling, low-cost, fully automatic vertical farm with a small footprint, suitable for the decreasing backyards in capital cities. In order to reduce cost, the design utilises a single set of sensors and tools moved by a robot arm. The authors believe that the concept presented herein may be the first complete solution based on the 'factory-in-a-box' concept. In addition, the proposed mechanism for self-assembly could potentially be used in other applications. This paper is organised as follows. Section 2 begins with a discussion on the requirements for a self-assembling vertical farm, followed by a detailed discussion of the proposed design to meet those requirements. Section 3 details the technical solutions, including dimensioning, a workspace analysis, and time studies for operation. Finally, Section 4 provides conclusions and ideas for future work.

## 2. A Self-assembling Vertical Farm

### 2.1. Requirements

The conceptual design presented in this paper targets autonomous vertical farming of vegetables. A prestudy led to the following objectives:

- Autonomous assembly, operation, and disassembly;
- Small footprint to suit an urban environment such as a small backyard;
- Centralised resources for cost reduction;
- Ability to be used both indoors and outdoors;
- Low power consumption;
- Ability to use solar energy;
- Integrated computer vision for phenotyping;
- Ability of greenhouse conversion.

### 2.2. Self-Assembly

As illustrated in Figure 1a, the proposed system would be delivered in two modules. The base module includes a base, the first stem, a robot arm, and growing platforms (leaves), while the resource module comprises stem expansion modules (SEMs), a power distribution system, water storage, and a battery pack. Both modules are wheeled and designed to be docked.

As illustrated in Figure 1b, when assembled, the farm has a tree-like structure with the base platform acting as the roots, the vertical guideway of the robot arm being the stem, and the growing platforms being the leaves.

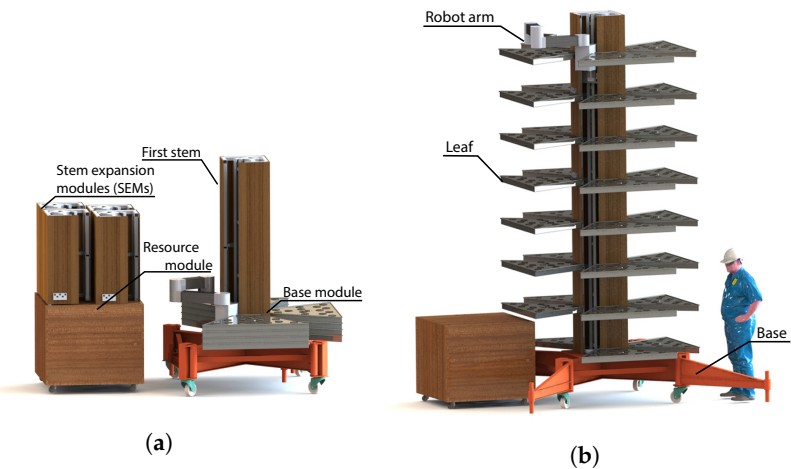

(**a**) (**b**)

**Figure 1.** A self-assembling autonomous vertical farm. Figure (**a**) includes a base module (right) and a resource module (left): (**a**) before assembly; (**b**) after assembly.

### 2.3. Self-Assembling Mechanism

The proposed system has the ability to assemble and disassemble itself with minimal user interactions as described below:

1 **Base expansion and levelling (Figure 2a,b)**
   The first stage of the installation involves moving the wheeled base module to the desired location and expanding the base. The base includes four stabilising legs, which are manually extended. Once extended, the support legs in the corners of the base and stabilising legs are adjusted to level the base. Once the base is fully extended, installed; and levelled, the weight of the system rests on eight support legs. The support legs include optional bolt-down brackets for outdoor use.

2 **Resource module docking (Figure 2c,d)**
   The resource module requires manual docking to the expanded base. Once the resource module is moved into the correct position, locks on two sides of the module

are engaged. After docking, the wiring harness and the water tube from the base module are connected to the resource module. Docking the station allows the robot arm on the base module to identify the exact position of the expansion modules.

3    **Power and water supply**

Single-phase power is connected to the power inlet on the resource module. A power connection is not required if using the optional solar panel solution, as the initial system setup would be conducted using the battery. A water supply is connected to the resource module, and the system will fill the tank automatically. The water tank also contributes to stabilise the system by lowering the centre of gravity.

4    **Initiating the system**

After booting the system, initial parameters must be provided, including the number of SEMs and the number of leaves. This is performed using the mobile app or web-based user interface by connecting to WiFi.

5    **Stem assembly (Figure 2e–m)**

The robot arm will reach the first SEM and secure it to the arm using the locking mechanism. Initially, the robot arm moves vertically to clear the resource module. Once cleared, the robot arm will position the SEM on top of the base stem section. Once the robot arm releases the lock that secures the SEM to the arm, a spring-loaded latching lock will activate and lock the SEM to the stem. Additional SEMs are installed following the same procedure.

6    **Leaf assembly (Figure 2n–t)**

A leaf is composed of a water catchment tray and a growing tray mounted on top of the water tray. The growing tray is delivered with pre-installed coconut fibre medium (coir) pellets. Coir is a low-cost growing medium, lighter than soil and can be compressed into pellets [25]. The coir pellets are replaced after a growing cycle is completed. The leaves are assembled by installing the uppermost layer first and then working downwards. Within a layer, the robot mounts the two leaves farthest from the robot vertical drive first. Thereafter, the remaining two leaves are installed before continuing with the next lower layer. The assembly order is designed to minimise interference with obstacles during the assembly. The vertical distance between leaves can be configured depending on the available light, plant height. and other relevant variables.

7    **Self-disassembling**

During a potential relocation, the robot would conduct a disassembly procedure where it would drain the leaves, remove leaves one by one and stack them, and finally remove the SEMs and stack them.

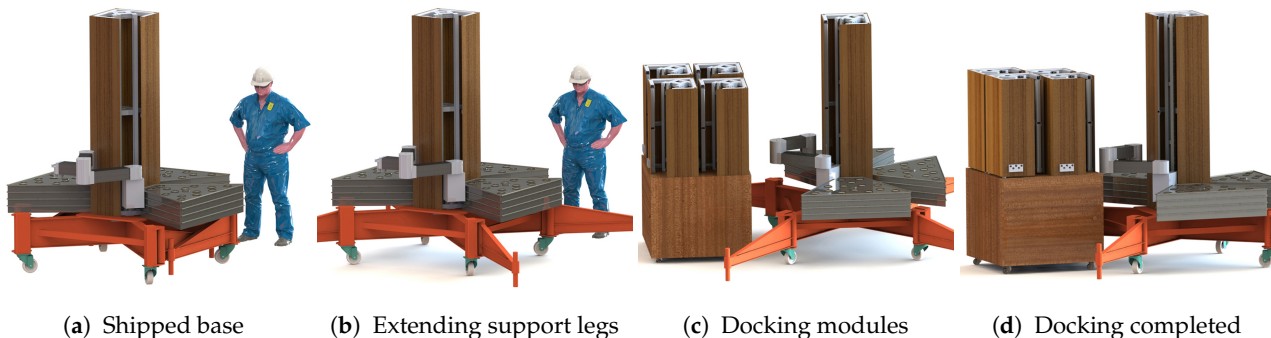

(**a**) Shipped base      (**b**) Extending support legs      (**c**) Docking modules      (**d**) Docking completed

**Figure 2.** *Cont.*

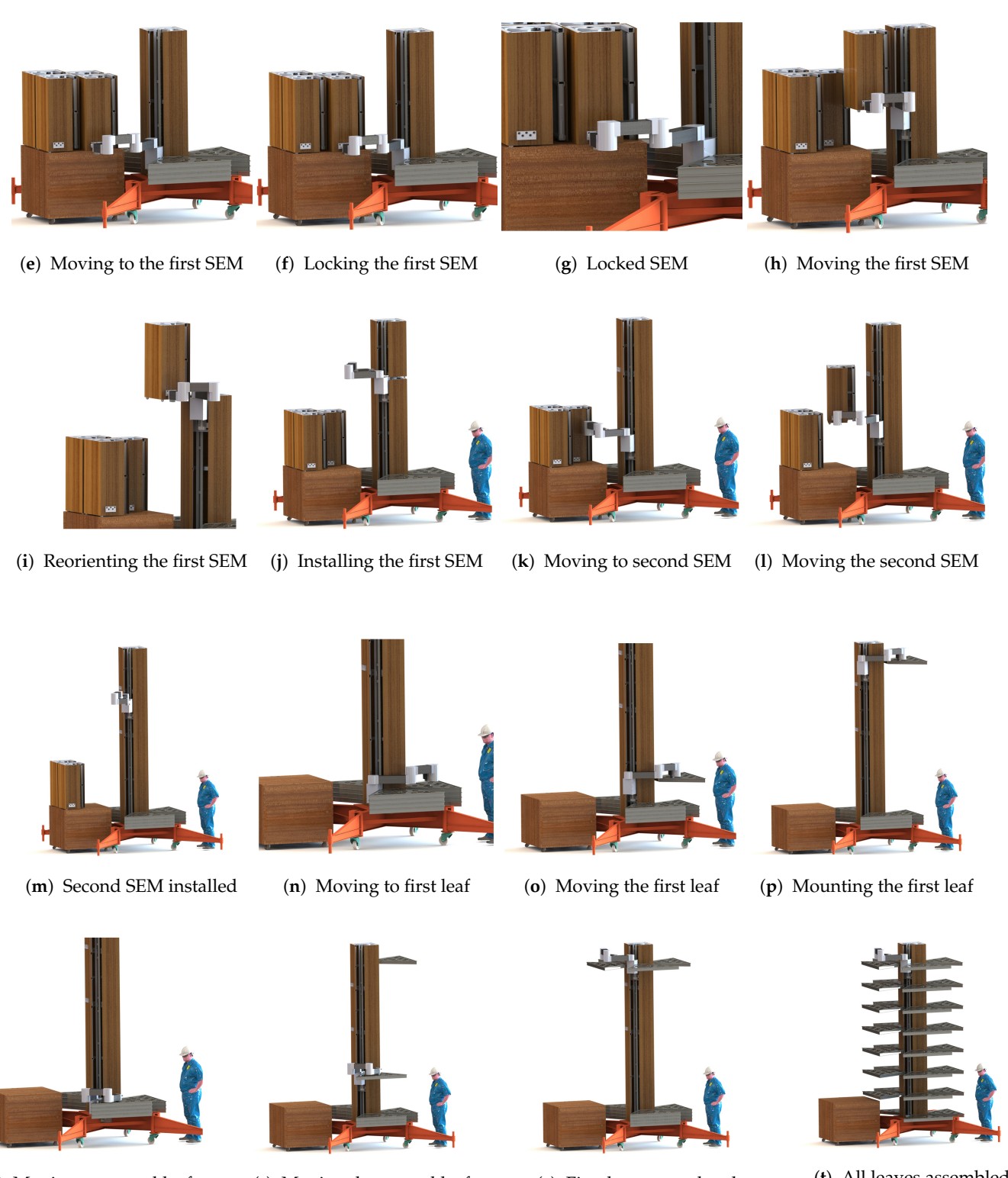

**Figure 2.** Stages of the self-assembly process.

*2.4. Functionality*

The descriptions below provide details of the system functionality:

- **Vertical motion of the robot arm**
  The structure is assembled and serviced by a robot arm that can position axis-symmetric tools in three Cartesian directions. The self-assembling feature is enabled by imple-

menting the vertical motion with a rack-and-pinion drive with bearing guides, The rack and bearing guides are part of both the base module and all SEMs, meaning that each added SEM increases the robot's vertical workspace.

- **Water and power ducting**
  The water and power supply duct design considers a spring-loaded winding duct mechanism that runs through the stem. This mechanism connects the water supply and the power supply from the resource module to the moving robotic arm.

- **Water supply**
  A water supply may be permanently connected to the resource module, or the built-in stock tank located inside the resource module may be manually filled. The high-pressure diaphragm pump inside the resource module delivers a measured amount of water to the tools mounted on the robot arm. The previously described greenhouse conversion is advised for optimum water usage.

- **Growing**
  The robot arm would continually service each leaf of the structure to perform seeding, watering, monitoring, fertilising, weeding, and other required operations for growing the plants. The growing trays or leaves are delivered with pre-installed coir pellets for growing plants. Once hydrated, the coir pellets expand, creating a rich soil-like growing base. The coir pellets are replaced manually before each growing cycle.

- **Harvesting**
  The vision system provides real-time information about the plants and continuous estimations on when the plants are ready to be harvested. Once a leaf is ready for harvest, the system would drain the bottom tray to remove excess water. Thereafter, the tray will be locked in and moved down onto the resource module for manual harvesting. After harvesting, the coir must be replaced before the leaf is remounted for the next growing cycle.

- **Growth optimisation**
  The solution is designed to monitor and adjust parameters continually. These parameters include plant density, water supply, sunlight exposure, plant tray positions, and potential fertiliser use. The objective is to optimise plant growth and allow the system to learn optimum growth parameters for different plant varieties and allocate resources accordingly. There may also be a possibility of modifying the growing parameters to delay the harvesting time if that would be required.

- **Self-powering capability**
  Section 3.4 shows the calculation of the power requirement and results from a solar panel compatibility analysis. The system is designed with the ability of harnessing energy from solar panels. The only modification required for this feature is to replace the top growing trays (leaves) with solar panels, as shown in Figure 3a. This does not affect the operations of the system except for reducing the growing area by one layer of leaves.

- **Greenhouse conversion**
  As shown in Figure 3b, the system may be converted into a greenhouse. Similar to the solar panel attachment, an umbrella-like structure is mounted on top of the structure, acting as a frame for a transparent membrane covering the entire tree.

- **IoT enabled system**
  The system is intended to share the optimum growth parameters between all installed systems. This information would be collected in a central database where any connected system may access optimum parameters to start the growing process. Hence, the set of optimal growing parameters would continually evolve.

- **Indoor-ready design**
  The system may be used indoors by providing the growing platforms (leaves) with light emitting diodes (LEDs). The LED panels may be installed underneath each leaf to provide the required light for the leaf below. In an indoor system, the uppermost leaves would not be used for growing plants.

- **Advanced agricultural research platform**
  The system monitors and stores growth data, which can be collected from multiple geographical locations with various environmental conditions. The controlled environment would help researchers understand and quantify parameters, such as the environmental impact on agriculture.

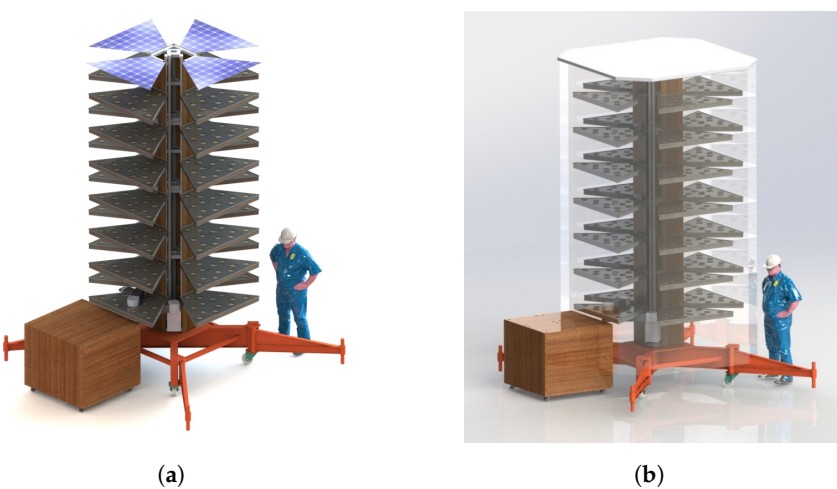

(**a**)                                                  (**b**)

**Figure 3.** Optional features: (**a**) inclusion of solar panels; (**b**) greenhouse conversion.

## 3. Technical Analysis

The mechanism has four degrees of freedom. Due to their low cost and ease of control, stepper motors were selected for both the prismatic joint and the three revolute joints of the robot arm. Worm and wheel gearboxes were selected for all joints to increase the detent torque and avoid using an electronic brake. This is particularly important for the prismatic joint as the robot should abort vertical motion in case of a power failure to prevent damaging the system.

### 3.1. Kinematics

As shown in Figure 4, a coordinate system is introduced with the origin in the centre of the stem at the lowest level the robot arm can travel. The axis of the first revolute joint of the robot arm is mounted at a distance of $l_0$ from the origin. This offset is introduced to increase the working angles of Joint 1 without interference between link $L_1$ and the stem. The last link in the chain is required in order to reach the stem with the orientation needed to mount the SEM.

The joint coordinate of the prismatic joint is denoted by $\alpha$, while the joint coordinates of the three revolute joints are denoted by $\theta_1, \theta_2$, and $\theta_3$ as specified in Figure 4. The coordinates of the TCP are denoted by x, y, z, and $\theta$ according to Figure 4. The forward kinematics can be expressed as:

$$\begin{cases} x = l_0 + l_1\,C(\theta_1) + l_2\,C(\theta_1 + \theta_2) + l_3\,C(\theta_1 + \theta_2 + \theta_3) \\ y = l_1\,S(\theta_1) + l_2\,S(\theta_1 + \theta_2) + l_3\,S(\theta_1 + \theta_2 + \theta_3) \\ z = \alpha \\ \theta = \theta_1 + \theta_2 + \theta_3 \end{cases} \tag{1}$$

where $C$ and $S$ denote the cosine and sine functions, respectively.

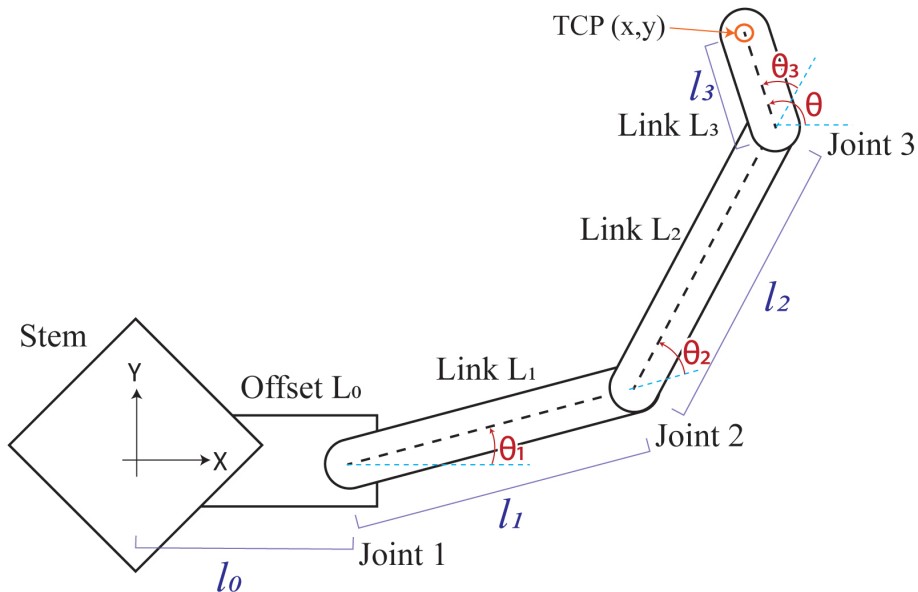

**Figure 4.** Kinematic parameters of the robot arm.

The joint angles of the manipulator are obtained from the TCP coordinates by first determining the coordinates of Joint 3 $(x_3, y_3)$ according to

$$\begin{cases} x_3 = x - l_3 \cos\theta \\ y_3 = y - l_3 \sin\theta \end{cases} \tag{2}$$

and thereafter utilising the theorem of cosines to calculate

$$\begin{cases} \theta_1 = \gamma - \beta & y \leqslant 0 \\ \\ \theta_1 = \gamma + \beta & y > 0 \end{cases} \tag{3}$$

where

$$\gamma = \text{atan2}\left( \frac{-y_3}{\sqrt{(x_3 + l_0)^2 + y_3^2}}, \frac{-(x_3 + l_0)}{\sqrt{(x_3 + l_0)^2 + y_3^2}} \right) \tag{4}$$

$$\beta = \text{acos}\left( \frac{-(x_3 + l_0)^2 - y_3^2 - l_1^2 + l_2^2}{2l_1 \sqrt{(x_3 + l_0)^2 + y_3^2}} \right) \tag{5}$$

The solution for $y \leqslant 0$ in Equation (3) corresponds to a left-arm solution, while the solution for $y > 0$ corresponds to a right-arm solution. Once a value of $\theta_1$ has been found, $\theta_2$ and $\theta_3$ are calculated according to

$$\begin{cases} \theta_2 = \text{atan2}\left( \dfrac{y_3 - l_1 S(\theta_1)}{l_2}, \dfrac{(x_3 + l_0) - l_1 C(\theta_1)}{l_2} \right) - \theta_1 \\ \\ \theta_3 = \theta - \theta_1 - \theta_2 \end{cases} \tag{6}$$

In order to avoid interference between link $L_1$ and the stem, a joint limit of $-135° < \theta_1 < 135°$ was introduced. Adjacent manipulator links operate in different planes and cannot interfere. However, in order to reduce the overall height of the robot arm, link $L_1$ and link $L_3$ operate in the same plane. To avoid interference between these links, a joint limit of $-165° < \theta_2 < 165°$ was derived using the collision check in Solidworks.

### 3.2. Link Length Selection and Workspace Analysis

The length $l_0$ was obtained using Solidworks collision check, identifying the minimum length required to avoid collision when link $L_1$ is parallel to the stem wall, as shown in Figure 5a. The length of link $L_3$ is the minimum length required to reach the SEM mounting orientation, as illustrated in Figure 5b.

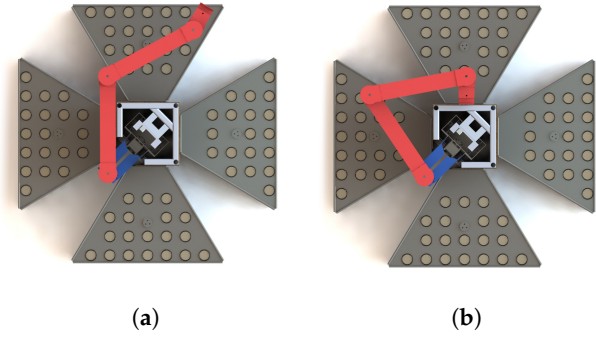

(**a**)                              (**b**)

**Figure 5.** Manipulator orientations (top view): (**a**) reaching the furthest point; (**b**) SEM mounting orientation.

In order to reach positions close to the stem, the lengths $l_1$ and $l_2$ were selected to be equal. As the mechanism is designed for axis-symmetric tools, this length was determined so that all positions on the leaves are reachable with at least one orientation. Workspace plots of the type shown in Figure 6 were calculated with increasing values of $l_1 = l_2$. The plot was obtained by evaluating all xy-positions in a grid using a step size of 0.1 m. Workspace positions that could be reached with at least one orientation $\theta$ are coloured in green.

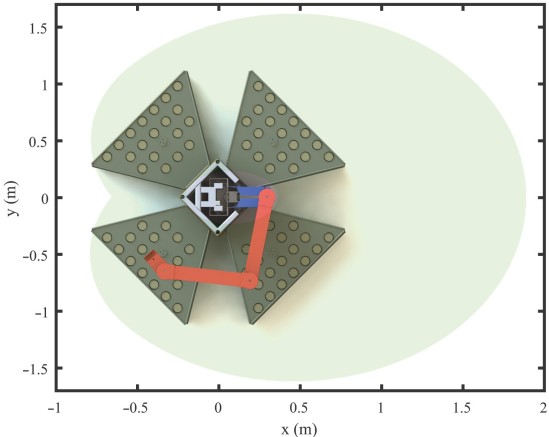

**Figure 6.** The horizontal workspace of the robot arm using the selected values of $l_0, l_1, l_2$, and $l_3$. The green area can be reached with at least one tool orientation.

The obtained values for the link lengths are $l_0$ = 500 mm, $l_1 = l_2$ = 600 mm, and $l_3$ = 120 mm.

### 3.3. Growth Area Analysis

The main advantage of vertical farming is the large arable area relative to the footprint. The footprint of the proposed structure is 9 m². This is the total area covered by the expanded legs without considering the optional bolted solution. The average arable area per leaf layer is 3.6 m². The base stem may include up to four leaf layers, whereas an SEM may include up to two leaf layers. Hence, with only two SEMs installed, the structure may include up to eight layers providing a total of 29 m² arable area, three times the footprint.

The number of SEMs per system depends on the use of optional bolted solution, indoor or outdoor use, ground support, and installation location.

### *3.4. Power Requirement Analysis*

In order to evaluate the potential of utilising solar panels, a power requirement analysis was performed. The analysis was based on the assumptions of an outdoor application without artificial lighting. In order to obtain some margin for the calculations, continuous operation of the components was assumed. It was estimated that components operate at 80% of the nominal power ratings and that the robot arm is perfectly horizontal, meaning there is no work against gravity during horizontal motions. The loss due to friction was not considered in the calculations. The mechanism was designed to use NEMA 17 stepper motors with a worm gear mechanism for revolute joints. The maximum continuous power consumption for a NEMA 17 stepper motor is 5 W [26]. A NEMA 23 stepper motor was selected for the prismatic joint. The maximum power consumption of the NEMA 23 motor is 108 W [27]. A high-pressure diaphragm pump with a power rating of 98 W is used to pump water from the tank to the tool. The control system power consumption was estimated assuming the use of a Raspberry Pi for controlling. As detailed in Table 1, the maximum power consumption for the system is 266 W.

**Table 1.** Estimated power use of the system.

|  | Rated (W) | Loss (%) | Actual (W) |
|---|---|---|---|
| Manipulator | 15 | 20% | 18 |
| Vertical drive | 108 | 20% | 130 |
| Water pump | 96 |  | 96 |
| Control system | 18 | 20% | 22 |
| Total maximum power consumption | | | 266 W |

In an outdoor configuration, the optional solar panels and the battery bank can be used to self-power the system. Considering a 15% efficient photovoltaic (PV) solar panel with a maximum power point tracker (MPPT) regulator, the average power output per square meter is 150 W [28]. Hence, at a minimum, the system requires $1.8\,\mathrm{m}^2$ of solar panels. A single layer of leaves covers $3.6\,\mathrm{m}^2$ and would produce twice the required power for operating the system. Additional power generated may be used as a backup power supply or may be connected to the power grid through a grid connect inverter.

### *3.5. Time Study*

A time study was conducted to identify the time required for the robot to service the tree and to determine achievable values of the Cartesian acceleration and maximum Cartesian speed used in the linear interpolation.

The simulation was based on a robot with eight layers of leaves, similar to Figure 5. The robot arm starts at the bottom of the tree and initially moves vertically to the uppermost layer. The TCP thereafter follows the path outlined in Figure 7 before moving to the next lower layer and repeating this process. The end configuration of the robot arm is the same as the start configuration. The acceleration and maximum speed used in the interpolation of the vertical motion were $0.2\,\mathrm{ms}^{-2}$ and $0.5\,\mathrm{ms}^{-1}$, respectively. These values are achievable with the proposed NEMA 23 stepper motor and transmission. For each leaf position, a reachable orientation was first determined, as close as possible to the orientation in the previous point. Starting from the previous orientation, $\theta$-angles were evaluated with an increment of $\pm 10$ degrees, and the first valid solution was used as the grid point orientation. The TCP follows a linear path between the points, starting and stopping at zero speed. Between each point, a delay of one second was introduced, corresponding to a seeding operation; however, this time would be highly dependent on the actual operation. As

shown in Table 2, the horizontal Cartesian interpolation between positions was performed using varying acceleration and maximum speed.

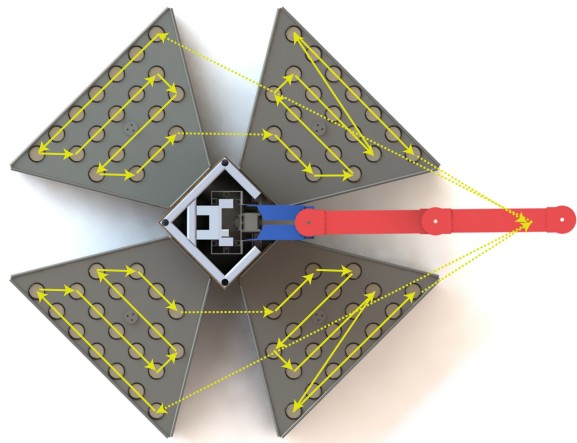

**Figure 7.** Robot path for servicing one layer of leaves.

**Table 2.** Time studies for different Cartesian accelerations and maximum speeds.

| Simulation | 1 | 2 | 3 | 4 |
|---|---|---|---|---|
| Max Cartesian speed ($ms^{-1}$) | 1.0 | 2.0 | 1.0 | 2.0 |
| Cartesian acceleration ($ms^{-2}$) | 1.0 | 1.0 | 2.0 | 2.0 |
| Max joint speed (°$s^{-1}$) [a] | 8.2 | 8.0 | 14.9 | 14.3 |
| Max joint acceleration (°$s^{-2}$) | 3.4 | 3.4 | 5.8 | 6.4 |
| Total time (s) | 1451.0 | 1449.0 | 1286.0 | 1106.0 |

[a] Calculated values.

### 3.6. Estimated Material Cost for a Prototype

Inverse kinematics were utilised to calculate the resulting joint speed and joint accelerations to verify if the selected Cartesian acceleration and maximum speed were achievable with the selected motors.

Table 2 includes the resulting execution time, maximum joint speed, and maximum joint acceleration for different Cartesian accelerations and maximum speeds when servicing a tree with eight layers of leaves according to the path in Figure 7.

The robot arm was designed to utilise NEMA 17 stepper motors with 1:15 gearing. Assuming continuous operation at 50% nominal speed, the maximum angular velocity is 9 °$s^{-1}$ and acceleration/deceleration is 6 °$s^{-2}$. The maximum angular velocity is calculated considering the maximum step frequency of the motor, gearbox speed reduction, and safety factors for joint rotation. Based on the simulation results, the selected Cartesian acceleration and maximum speed were 2 $ms^{-2}$ and 1 $ms^{-1}$, respectively.

The simulation results indicate that the effect of maximum velocity on the total time is negligible. This is expected as the majority of motion segments are too short for the robot to reach maximum speed. However, the effect of the used acceleration is considerable and selecting motors with a high acceleration is an important design consideration. As shown in Table 2, with the selected values of acceleration and maximum speed, the robot takes nearly 21 min to service a structure of eight layers.

Table 3 provides an estimation of the material cost for a prototype. At the time of publication, the cost for a standard configuration is estimated about AUD 10,700. It should be noted that the unit price for a one-off prototype is significantly higher compared to the cost at mass production. As a comparison, a container farm is priced in between AUD 20,000 and AUD 25,000 [29].

**Table 3.** Cost estimation for prototyping the self-assembling robotic farm.

| Segment | Rate (AUD) | Qty | Cost (AUD) |
|---|---|---|---|
| Base | 1000 | 1 | 1000 |
| Base stem* | 1500 | 1 | 1500 |
| Robot arm including tools | 1000 | 1 | 1000 |
| Control system | 600 | 1 | 600 |
| Resource module | 1000 | 1 | 1000 |
| SEM (per module) | 800 | 2 | 1600 |
| growing platform (per leaf) | 200 | 20 | 4000 |
| LED grow lights [#] | 1000 | 20 | 20,000 |
| Solar panels and battery [#] | 4500 | 1 | 4500 |
| Total of the standard (non optional) components | | | 10,700 |

* The base stem includes a vertical drive, [#] Optional.

## 4. Conclusions

This paper introduced a conceptual architecture for a self-assembling vertical farm with a small footprint, designed for urban use. It is designed for autonomous farming by non-specialist users. The main novelty is the self-assembling mechanism which would be useful in any application where a temporary vertical structure must be assembled and serviced by a robot arm.

The computer-aided design was verified by mathematical modelling, confirming the ability of the robotic arm to reach all plants with at least one tool orientation. It was also verified that all operations of the robot can be carried out without mechanical interference. A power usage estimation confirms the viability of the solar-powered self-sustaining system. The proposed robotic farm has the potential to be a cost-effective solution compared to similar farming solutions. An array of the proposed mechanisms might be useful in commercial-scale agriculture. As the architecture is designed for both outdoor and indoor usage, an array of the proposed mechanisms may also be operated in a factory setting.

Although the presented design is only conceptual, effort has been made to theoretically evaluate the practical feasibility of various components.

*Prospects for Further Research*

Planned future research includes building a functional prototype for real-world evaluation. Further work should also include detailed studies on cultivation-related factors, such as cycle time, soil requirements, plant height, plant weight, and plant width and density which are essential to fully evaluate the proposed concept. The system can be further developed to tilt the leaf structures for sun tracking to increase light utilisation, adding a robotic platform for self-positioning and automatic docking functionality.

**Author Contributions:** Conceptualization and methodology, B.W and M.I.; software, B.W; validation, B.W and M.I.; formal analysis, B.W and M.I.; data curation, B.W; writing—original draft preparation, B.W; writing—review and editing, M.I.; visualization, B.W.; supervision, M.I.; project administration, M.I. All authors have read and agreed to the published version of the manuscript.

**Funding:** This research received no external funding.

**Institutional Review Board Statement:** Not applicable.

**Informed Consent Statement:** Not applicable.

**Data Availability Statement:** Not applicable.

**Conflicts of Interest:** The authors declare no conflict of interest.

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
