# Peer review of "Design and Simulations of a Self-Assembling Autonomous Vertical Farm for Urban Farming"

_agriculture, doi:10.3390/agriculture13010112_

Round 1

Reviewer 1 Report

Vertical farms have been widely concerned in recent years, so this article is significant and interesting. However, many contents of the article are lack of standardization.

1.There is no clear core problem to be solved as a goal of the article, so the whole article is just a description of equipment design.

2.Low cost and automation were mentioned in line 86, however these two aspects are contradictory, and no further analysis was made in the article.

3.The Centralised resources for cost reduction in line 102 and Integrated computer vision for phenotyping in line 106 as the design targets were not touched in the design in the article.

4.The cultivation and plant were not considered in the design targets. Such cultivation cycle time, cultivation tray type, plant height, plant weight, plant width and cultivation density should be focused on.However, only electromechanical factors were considered in the article.

5.It is very important for a article publishing in the Agriculturethat the cultivation step of the self-assembling autonomous vertical farm should be explained. But the design was explained mainly.

6. There are no explains of x3, y3,γ and β in 3.1.

7. The basis of the simulation results in Table 2 is not clear in 3.5 Time study .

8. Many functions were explained in 2.4. Functionality. However, it is confused to understand the aim of the article.

9.Dismounting way of water supply duct and fixing way of LED were not explained in the article.

10.The article introduced a conceptual design of an autonomous vertical farm, but it was not compared with other types of vertical farm such as scissor lift and logistic equipment.

11. The article is lack of additional experiments to support its conclusion, and the conclusion format of the article is not standardized.

12.For a vertical farm in the article all step of seeding, transplanting, delivering, cultivating and washing should be considered which could build a cultivation system. And the feasibility of integration between cultivation mode and the Self-Assembling Autonomous Vertical Farm is not clear.

13. Although the title of the article includes simulation,there is little simulation content in the article.

14. 

Author Response

Response to Reviewer 1

Vertical farms have been widely concerned in recent years, so this article is significant and interesting.

Thanks for the nice comment.

However, many contents of the article are lack of standardization.

We have attempted to address your comments below.

  • There is no clear core problem to be solved as a goal of the article, so the whole article is just a description of equipment design.

The core problem we identified is discussed in the introduction section. The introduction discusses the limitations of current automation solutions for home farming including a large required growing area which may not be available in an urban environment and the need for a solution that does not require technical or agricultural know-how.

  • Low cost and automation were mentioned in line 86, however these two aspects are contradictory, and no further analysis was made in the article.

We included a cost calculation in the initial iterations of the paper as a rough financial estimation; however, the calculations were omitted in the final version to improve publication space utilisation. Based on your comment, we have reintroduced this section. (Section 3.6)

  • The Centralised resources for cost reduction in line 102 and Integrated computer vision for phenotyping in line 106 as the design targets were not touched in the design in the article.

The entire design is based on the concept of centralising the resources to reduce the cost. The centralisation of resources is the use of a robotic arm that carries a single set of sensors and actuators instead of having such resources at each plant. Based on your comment, we have clarified this by adding the sentence “The system is designed for cost minimisation, using one set of resources moved by a robot arm to service the plants.”

As expressed in the abstract and introduction, the focus of this paper is to introduce a conceptual design of an autonomous vertical farm where the main novelty is the self-assembling feature. As you rightly point out, computer vision has not been touched on in this manuscript as such a wide topic requires a dedicated paper. We are currently finalising a paper on using depth cameras and machine learning for phenotyping of microgreens.

  • The cultivation and plant were not considered in the design targets. Such cultivation cycle time, cultivation tray type, plant height, plant weight, plant width and cultivation density should be focused on. However, only electromechanical factors were considered in the article.

We certainly agree that these elements are missing in the current article. As you point out, this introductory article mainly focuses on the electromechanical factors and significant further work is required to address the important design target you mention. In response to your comment, we have included a sentence about this in the conclusion section: “Further work should also include detailed studies on cultivation-related factors such as cycle time, soil requirements, plant height, plant weight, plant width and density which are essential to fully evaluate the proposed concept”

  • It is very important for a article publishing in the “Agriculture” that the cultivation step of the self-assembling autonomous vertical farm should be explained. But the design was explained mainly.

We can certainly see your point of view and this article would normally not have been submitted to ‘Agriculture’, however, as the article was submitted to the special issue “Advances in Agricultural Engineering Technologies and Application” we believe that this allows a larger variation in the type of articles published. 

  • There are no explains of x3, y3,γ and β in 3.1.

We noticed that some equation numbers were missing which we have now added. After this change, we refer to the text before equation (1) which states “the coordinates of joint 3 (x3,y3)” and that the definitions of y and β are provided in (4).

  • The basis of the simulation results in Table 2 is not clear in 3.5 Time study.

Of the parameters in Table 2, only the maximum speed and acceleration are selected values, the joint values are calculated from the selected values by applying the inverse kinematics. The reason for selecting cartesian maximum speed and acceleration in the range 1-2 is that we have considered the optimum speeds and accelerations of the selected motors and gear mechanisms and the safety factors for the joint rotation as mentioned in lines 333 to 336.  The inverse calculation shows if the selected values are possible to achieve. Based on your comment, we have tried to clarify this section. 

  • Many functions were explained in 2.4. Functionality. However, it is confused to understand the aim of the article.

We appreciate that the article may seem confusing as it deviates from the typical article in ‘Agriculture’. As previously stated, the aim of the article was to introduce a new conceptual design of a self-assembling vertical farm and analyse some of its properties. We certainly acknowledge that this is the first paper and much more work is required.

  • Dismounting way of water supply duct and fixing way of LED were not explained in the article.

Thank you for your comment. We have revised the section and included details on the water supply and power supply mechanism. Both the water and power supply are connected to the robotic arm and are not affected by the self-assembling mechanism.

“Water and power ducting

The water and power supply duct design considers a spring-loaded winding duct mechanism that runs through the stem. This mechanism connects the water supply and the power supply from the resource module to the moving robotic arm.“

  • The article introduced a conceptual design of an autonomous vertical farm, but it was not compared with other types of vertical farm such as scissor lift and logistic equipment.

We have not seen any fully automated vertical farm. The examples we have seen using scissor lifts have not been part of an automation solution but used for manual harvesting, e.g as described here https://www.accessinternational.media/news/Genie-scissors-assist-Europe-s-largest-vertical-farm/8014093.article

  • The article is lack of additional experiments to support its conclusion, and the conclusion format of the article is not standardized.

Thanks for pointing this out. We have familiarized ourselves more with the format of this journal and rewritten the conclusion section.

  • For a vertical farm in the article all step of seeding, transplanting, delivering, cultivating and washing should be considered which could build a cultivation system. And the feasibility of integration between cultivation mode and the Self-Assembling Autonomous Vertical Farm is not clear.

Thank you for your comment. As previously mentioned, the article focuses on the Design and Simulations of a Self-Assembling Autonomous Vertical Farm where the key focus is the self-assembling capability. We agree that there are several other topics that must be explored to evaluate the practical feasibility of this concept. We have outlined some of this further work in the conclusion section.

  • Although the title of the article includes simulation there is little simulation content in the article.

Simulations include design simulations in CAD and mathematical modelling to determine suitable acceleration and maximum speed in addition to a time-study simulation.

Reviewer 2 Report

General:

Congratulations for complete Interdisciplinary paper! The article is interesting, the topic is actual and results useful!

The manuscript brings contributions in urban smart agriculture topic, by integrating vertical farming systems with a smart solution for monitoring the automated cultivation process in household and portable farming.

Generally, I agree on publication.

However, there are some issues that must be addressed to improve the overall communication of author`s work:

·       Please, insist on actual stage in vertical farming (more references) based of the paper

·       Some parts need to be re-systematized.

Introduction

Analyze is clear, but, sometimes too comprehensive to the field. Must be presented relevant aspects in correlation with title and objectives.

Objective can be more expressive relative to title and content (Rows 86-88). Rows 88-90, please justify according to other references.

A Self-assembling Vertical Farm

Fig 1 and fig 2, it is better to indicate the most important parts for avoid confusions of the readers.

Fig 4 can be summarized.

The readers should find here (only and exclusively) detailed description of all your concept (step by step).

Technical Analysis

Table 2 and other text content (ex. row 331): please, use the multiplication for measurement units (like in row 334).

Conclusion

Conclusions must be consistent with the evidence and arguments presented. Please provide comparations with other results that are limited to your case study (use references in the same frame of results).

Author Response

Response to Reviewer 2

Congratulations for complete Interdisciplinary paper! The article is interesting, the topic is actual and results useful! The manuscript brings contributions in urban smart agriculture topic, by integrating vertical farming systems with a smart solution for monitoring the automated cultivation process in household and portable farming. Generally, I agree on publication. However, there are some issues that must be addressed to improve the overall communication of author`s work:

Thank you

  • Please, insist on actual stage in vertical farming (more references) based of the paper. Some parts need to be re-systematized.

Thank you for your comment. In response to your comment, we have reviewed the introduction section, rearranged it, and re-systematized the conclusion section as described in the response for comments 2,3 and 5.

  • Analyze is clear, but, sometimes too comprehensive to the field. Must be presented relevant aspects in correlation with title and objectives. Objective can be more expressive relative to title and content (Rows 86-88). Rows 88-90, please justify according to other references.

Thank you for your comment. Appreciate your thoughtfulness towards justification of the claim ‘the first complete solution based on ‘Factory-in-a-box’ concept’. We have reviewed the section and included a justification for the claim with relevant references.

“The self-assembling vertical farm solution presented in this paper can be considered as the first step towards a ‘Robot-based self-assembling mobile factory’, a novel concept, where the factory can assemble itself, automate the process with the same resources used for assembly, and when required, dismantle and pack itself for transportation. Concepts such as ‘Factory-in-a-box’ [16] and ‘Mobile Manufacturing System’ [17,18] have been discussed in academia since 2007.”

The following references were added:

  • Jackson, M.; Zaman, A. Factory-In-a-Box-Mobile Production Capacity on Demand. Technical Report 1, 2007
  • Stillström, C.; Jackson, M. The concept of mobile manufacturing. Journal of Manufacturing Systems 2007, 26, 188–193. https://doi.org/10.1016/j.jmsy.2008.03.002.
  • Benama, Y.; Alix, T.; Perry, N. Framework definition for the design of a mobile manufacturing system. Advances on Mechanics, Design Engineering and Manufacturing 2017. https://doi.org/10.1007/978-3-319-45781-9{_}.

      We have also cut out a significant part of the background to modular robotics in an effort to address the “too comprehensive” issue and better fit this journal.

  • Fig 1 and fig 2, it is better to indicate the most important parts for avoid confusions of the readers. Fig 4 can be summarized. The readers should find here (only and exclusively) detailed description of all your concept (step by step).

Thank you again for making this effort to review our paper in detail, much appreciated. We have included labels in the figure.

<Please refer PDF for the image>

We believe that Fig. 4 images indicate crucial steps in the assembly process and the reader might get confused without them.

  • Table 2 and other text content (ex. row 331): please, use the multiplication for measurement units (like in row 334).

Thank you for pointing this out. We have now corrected the measurement units.

  • Conclusions must be consistent with the evidence and arguments presented. Please provide comparations with other results that are limited to your case study (use references in the same frame of results).

We have completely revised the conclusion section carefully limiting it to the results that were actually derived in the paper

Reviewer 3 Report

The article presents a conceptual model of an urban Self-Assembling Autonomous Vertical Farm. The article is well illustrated, provides a detailed description of the design and parameters of the work. To improve the quality of the article, the authors need to consider the following points for revision.

1.) It would be interesting to present a design calculation of the economic efficiency of the proposed Self-Assembling Autonomous Vertical Farm.

2). To improve the quality of work, you should also add the section "Prospects for Further Research".

3.) In the prospects of further research, the possibility of automating the process of docking the resource module should be considered, for example, using robotic platforms for transporting modules without human intervention.

4.) In the section "1. Introduction", after analyzing the existing vertical trusses, it is necessary to describe in more detail the disadvantages and advantages of existing solutions, including their designs.

5.) In the section "4. Conclusion" it would be logical to briefly describe the final technical characteristics of the vertical truss, possibly in tabular form.

6.) The section "4. Conclusion" should also clearly describe the novelty of the proposed conceptual model.

Author Response

Response to Reviewer 3

The article presents a conceptual model of an urban Self-Assembling Autonomous Vertical Farm. The article is well illustrated, provides a detailed description of the design and parameters of the work. To improve the quality of the article, the authors need to consider the following points for revision.

Thank you for your suggestions which we have attempted to address.

  • It would be interesting to present a design calculation of the economic efficiency of the proposed Self-Assembling Autonomous Vertical Farm.

This is a really good idea. We included a cost calculation in the initial iterations of the paper as a rough financial estimation. Based on the reviewer’s comments, we have reintroduced such a section. (Section 3.6)

  • To improve the quality of work, you should also add the section "Prospects for Further Research".

Thank you for your comment. In response to your comment, we have introduced the ‘Prospects for Further Research’ under the conclusion section.

  • In the prospects of further research, the possibility of automating the process of docking the resource module should be considered, for example, using robotic platforms for transporting modules without human intervention.

Thanks for the suggestion. We have added a comment in the revised conclusion section.

“Planned future research includes building a functional prototype for real-world evaluation. Further work should also include detailed studies on cultivation-related factors such as cycle time, soil requirements, plant height, plant weight, plant width and density which are essential to fully evaluate the proposed concept. The system can further be developed to tilt the leaf structures for sun tracking to increase light utilisation, adding a robotic platform for self-positioning, and automatic docking functionality.”

  • In the section "1. Introduction", after analyzing the existing vertical trusses, it is necessary to describe in more detail the disadvantages and advantages of existing solutions, including their designs.

Thank you for your suggestion. As you have commented, it would be better to discuss the advantages and the disadvantages of the existing solutions. We have included the following in response to the comment.

“The expansion mechanism of a tower crane is straightforward, however, in contrast to the mechanism proposed in this paper, a tower crane is not designed to access its own base structure.”

  • In the section "4. Conclusion" it would be logical to briefly describe the final technical characteristics of the vertical truss, possibly in tabular form.

Thank you for your comment. In response to the comment, we have revised the conclusion section and included the following section describing the results obtained.

“The computer-aided design was verified by mathematical modelling, confirming the ability of the robotic arm to reach all plants with at least one tool orientation. It was also verified that all operations of the robot can be carried out without mechanical interference. A power usage estimation confirms the viability of the solar-powered self-sustaining system. The proposed robotic farm has the potential to be a cost-effective solution compared to similar farming solutions.”

  • The section "4. Conclusion" should also clearly describe the novelty of the proposed conceptual model.

Thank you for your comment. In response to the comment, we have revised the conclusion section by adding the following description.

“This paper introduced a conceptual architecture for a self-assembling vertical farm with a small footprint, designed for urban use. It is designed for autonomous farming by non-specialist users. The main novelty is the self-assembling mechanism which would be useful in any application where a temporary vertical structure must be assembled and serviced by a robot arm.

Round 2

Reviewer 1 Report

I still think that this manuscript is not in line with the direction of "Agricultural" .

1. This manuscript focused on how to assemble automatically and gives some equipment parameters, which should be published in mechanical engineering magazines.

2. For publishing in "Agriculture" , it should focus on how to operate the equipment based on the cultivation process.

3. Some technical requirements were mentioned from line 98 to line 104,however the requirements were not disscused in the manuscript  such as small footprint, centralised resources for cost reduction, low power consumption. And how to solve the contradiction between using solar energy and plants needing light?

Author Response

As there are no details on how we can change the papers without completely rewriting it which is not possible, we leave it to the editor to accept or reject the paper.